# Analysis of Lsm Protein-Mediated Regulation in the Haloarchaeon *Haloferax mediterranei*

**DOI:** 10.3390/ijms25010580

**Published:** 2024-01-01

**Authors:** Gloria Payá, Vanesa Bautista, Sandra Pastor-Soler, Mónica Camacho, Julia Esclapez, María-José Bonete

**Affiliations:** Department of Biochemistry and Molecular Biology and Soil Science and Agricultural Chemistry, Faculty of Science, University of Alicante, Ap 99, 03080 Alicante, Spain; gloria.paya@ua.es (G.P.); vanesa.bautista@ua.es (V.B.); sandra.pastor@ua.es (S.P.-S.); camacho@ua.es (M.C.); julia.esclapez@ua.es (J.E.)

**Keywords:** like-Sm proteins, Hfq, Lsm, nutritional stress conditions, microarray

## Abstract

The Sm protein superfamily includes Sm, like-Sm (Lsm), and Hfq found in the *Eukarya*, *Archaea*, and *Bacteria* domains. Archaeal Lsm proteins have been shown to bind sRNAs and are probably involved in various cellular processes, suggesting a similar function in regulating sRNAs by Hfq in bacteria. Moreover, archaeal Lsm proteins probably represent the ancestral Lsm domain from which eukaryotic Sm proteins have evolved. In this work, *Haloferax mediterranei* was used as a model organism because it has been widely used to investigate the nitrogen cycle and its regulation in Haloarchaea. Predicting this protein’s secondary and tertiary structures has resulted in a three-dimensional model like the solved Lsm protein structure of *Archaeoglobus fulgidus*. To obtain information on the oligomerization state of the protein, homologous overexpression and purification by means of molecular exclusion chromatography have been performed. The results show that this protein can form hexameric complexes, which can aggregate into 6 or 12 hexameric rings depending on the NaCl concentration and without RNA. In addition, the study of transcriptional expression via microarrays has allowed us to obtain the target genes regulated by the Lsm protein under nutritional stress conditions: nitrogen or carbon starvation. Microarray analysis has shown the first universal stress proteins (USP) in this microorganism that mediate survival in situations of nitrogen deficiency.

## 1. Introduction

The Sm protein superfamily includes Sm and like-Sm (Lsm), found in the *Eukarya* and *Archaea* domains, respectively, and Hfq proteins, found in the *Bacteria* domain, and only one archaeon species, *Methanocaldococcus jannaschii* [1,2,3,4,5]. The Sm and Lsm proteins differ at the amino acid sequence level from Hfq; however, they show striking similarities in their tertiary and quaternary structure levels [3,6,7]. The Sm protein superfamily has a bipartite sequence known as the Sm motif, which consists of two segments, the Sm1 and Sm2 motifs, separated by a region of variable amino acid sequence and length [7]. The secondary structure of this protein superfamily consists of a short α-helix (two to four turns) at the N-terminal end and five β-strands, identified from the N-terminal to the C-terminal end as β1, β2, β3, β4, and β5. The Sm1 motif corresponds to the β1, β2, and β3 strands, and the Sm2 motif corresponds to the β4 and β5 strands [8].

Bacterial Hfq proteins form stable hexamers that preferentially bind to A-U-rich sequences [9], whereas eukaryotic Lsm proteins form up to five heteroheptameric complexes. In contrast, the structures of archaeal Lsm proteins are composed of homohexameric [10] or homoheptameric complexes [8,11,12,13,14]. The hexameric or heptameric rings of the Hfq and Sm/Lsm proteins consist of Lsm subunits that assemble without RNA [8,12,13]. In *Archaeoglobus fulgidus*, the Lsm2 protein was found to form hexamers without RNA but was shown to assemble into stable heptamers upon adding U-rich RNA [10,15]. Moreover, in *Haloferax volcanii,* it was determined via Laser-Induced Liquid Bead Ion Desorption Mass Spectrometry (LILBID-MS) that the Lsm protein heterologously overexpressed in *Escherichia coli* forms homoheptameric complexes in vitro, and masses corresponding to the Lsm monomers, dimers, trimers, and tetramers were also observed [16].

Numerous studies have shown that eukaryotic Sm proteins function as molecular scaffolds for RNP assembly and are involved in mRNA degradation, mRNA processing, mRNA stabilization, telomere maintenance, and histone maturation [7,17,18,19]. Archaeal Lsm proteins were discovered by means of sequence homology in the database search [20]. These proteins were not necessarily expected in archaea due to the absence of introns in their genes and their primitive RNA processing machinery [21]. On the other hand, bacterial Hfq proteins have many cellular functions [6]. For example, one of the identified functions of Hfq corresponds to its interaction with the sRNAs that regulate gene expression, cooperating as an RNA chaperone by facilitating the binding of regulatory sRNAs to their target mRNAs [22]. Archaeal Lsm proteins have been shown to bind sRNAs. They are probably involved in various cellular processes, suggesting a similar function in regulating sRNAs by Hfq in bacteria [16,23,24]. Archaeal Lsm proteins probably represent the ancestral Lsm domain from which eukaryotic Sm proteins have evolved [12].

Moreover, deletion mutants of the *hfq* gene have been studied in several bacterial species, showing pleiotropic phenotypes where the growth rate, cell morphology, and tolerance to stress conditions are affected. Specifically, the Hfq proteins regulate stress conditions such as heat shock, oxidative stress, osmotic stress, UV exposure, acidic pH, and ethanol stress [25,26,27,28,29,30]. In addition, they play an essential role in the motility and virulence of pathogenic bacteria [3,31]. Deletion mutants of the *lsm* gene have been successfully generated in two haloarchaeal species, specifically in *Hfx. volcanii* [16,32] and *Haloferax mediterranei* [33]. In *Hfx. volcanii,* the *lsm* gene cotranscribes with the overlapping gene encoding the ribosomal protein L37r [16]. Moreover, a deletion mutant of the Sm1 motif versus the parental strain showed differences in the transcriptome in the rich medium (Hv-YPC), and the deletion mutant showed increased motility activity compared to the parental strain [33]. In *Hfx. mediterranei*, the *lsm* deletion mutant and the Sm1 motif deletion mutant were generated and characterized. A comparison of the deletion mutant and the parental strain HM26 under standard and stress growth conditions revealed differences in growth. These results indicate that the Lsm protein is involved in standard and stress growth conditions (low/high salinity, low/high temperature, heat shock, oxidative stress, and ethanol stress). Furthermore, expression of the *lsm* and *rpl37e* genes was constitutive, and the cotranscription of both genes occurs at suboptimal salt concentrations and temperatures [33].

A recent bioinformatics study of the distribution of the Lsm proteins in the *Archaea* domain showed that most species of the phylum Euryarchaeota present only one Lsm protein, while most species of the phylum Crenarchaeota present two Lsm proteins. Many of these genes are adjacent to the transcriptional regulators of the Lrp/AsnC and MarR families, RNA-binding proteins, and ribosomal protein L37e. Notably, only proteins from species of the class Halobacteria conserved the internal and external residues of the RNA-binding site identified in *Pyrococcus abyssi*. Finally, in most species, the *lsm* genes show associations with genes that encode proteins closely related to RNA metabolism [34]. The role of Lsm in *Archaea* remains unknown; however, they appear to play a crucial role in RNA metabolism. Therefore, more work is needed to elucidate if Lsm proteins, e.g., act as chaperones that facilitate the folding of sRNAs as in bacteria and/or act as structural scaffolds for the assembly of RNPs as in eukaryotes [21]. Recently, it has been reported that SmAP1 or Lsm binding to RNA is one of the mechanisms of post-transcriptional regulation in *Halobacterium salinarum* [35].

Methods for producing and purifying proteins from haloarchaea are essential for structural studies and further biotechnological applications. The use of *E. coli* as a host for the overexpression of halophilic proteins has several limitations due to the high content of acidic amino acid residues on the surface of these proteins and the requirement of a high salt concentration for their correct folding [36,37,38]. Homologous overexpression and purification of proteins under native conditions have been optimized in *Hfx. volcanii* [39,40,41]. The enzyme nitrite reductase (NirK) from *Hfx. mediterranei* has been overexpressed in *Hfx. volcanii,* as a halophilic host, which allowed this enzyme’s purification and characterization [42]. In addition, the first work on the homologous overexpression and purification of the regulatory protein Lrp from *Hfx. mediterranei* has recently been published [43].

In this work, *Hfx. mediterranei* was used as a model organism because it has been widely used to investigate the nitrogen cycle and its regulation in Haloarchaea. The homologous overexpression and purification via molecular exclusion chromatography have allowed us to show the aggregation state of the protein. The structural bioinformatic analysis of the Lsm protein has been performed to obtain 3D models. In addition, the effect of the mutation of the Sm1 motif (HM26-Δ*Sm1*) in conditions of nutrient deficiency has been analyzed using microarrays and compared with the parental strain (HM26). The study of transcriptional expression has allowed us to obtain the target genes regulated by the Lsm protein under nutritional stress conditions such as nitrogen or carbon starvation. This work increases the general knowledge about the function of the Lsm proteins in the *Archaea* domain and *Hfx. mediterranei* in particular.

## 2. Results

### 2.1. Bioinformatic Analysis of Haloferax mediterranei Lsm Protein

#### 2.1.1. In Silico Analysis at the Primary and Secondary Structure Levels

The percentage of similarity between the Lsm protein of *Hfx. mediterranei* and homologous proteins of eukaryotic species is higher than 45%, despite having a high percentage of acidic residues. These results agree with Payá et al. 2021 [33], as the amino acid sequence of Lsm from *Hfx. mediterranei* corresponds precisely to the consensus amino acid sequence of eukaryotic Lsm proteins.

The Lsm protein structure of *P. abyssi* is a heptameric ring with a central cavity, like eukaryotic Sm proteins. RNA molecules bind to the protein at two different sites: within the ring with three residues defining the uridine-binding pocket and on the surface of the α-helix located in the N-terminal region. The internal uracil-binding pocket is formed by the residues His-37, Asn-39, and Arg-63. The uracil base establishes contacts with His-37 and Arg-63. The binding pocket is stabilized by a salt bridge between Arg-63 and Asp-65, forming an ionic interaction with Lys-22. In addition, the hydrogen bonds at Asp-35 and Asn-39 make this binding site specific for uridine. The external RNA-binding site residues in the α-helix are Arg-4, Asp-7, His-10, and Tyr-34 [44]. *Hfx. mediterranei* retains all the RNA-binding residues of the internal and external pockets identified in *P. abyssi*. These residues are exclusively conserved in the Lsm proteins of Halobacteria class species [34].

The secondary structure of the *Hfx. mediterranei* Lsm protein was predicted using Sequence Annotated by Structure, SAS [45], and Jpred 4 [46]. An α-helix is obtained at the N-terminal end, followed by six β-sheets using both tools (Figure 1). However, the β-sheets named B3–B5 differ depending on the tool used (Figure 1A,B). Since the Lsm proteins are characterized by an α-helix at the N-terminal end, followed by five β-strands [21], the most likely hypothetical structure is the one proposed in Figure 1C.

#### 2.1.2. Homology Modeling

Lsm protein alignment based on the amino acid sequence against the PDB database [47] was performed using the BLAST tool [48] to obtain a suitable reference structure for modeling the protein of interest. The Lsm proteins from *Hbt. salinarum*, *A. fulgidus*, and *P. abyssi* species were selected as reference structures (Appendix A). The Lsm protein from *Hbt. salinarum* R1, with a size of 62 amino acids (accession number: 6TFL_A), showed the highest coverage (98%) and sequence identity (62.67%), as well as the lowest *E*-value (1 × 10^−^^23^); the Lsm protein from *A. fulgidus*, with a size of 77 amino acids (accession number: 1I4K_1), showed 97% coverage and 45.95% sequence identity, with an *E*-value of 1 × 10^−^^11^; and the Lsm protein of *P. abyssi*, with a size of 75 amino acids (accession number: 1H64_1), showed 97% coverage and 45.95% sequence identity, with an *E*-value (9 × 10^−^^13^).

Sequence homology modeling employing SWISS-MODEL was carried out using the three reference structures mentioned above, obtaining the parameters in Appendix A and the structures in Figure 2. The residues with a lower confidence index correspond to those between the β3 and β4 sheets (Figure 2A). The structure obtained using the *A. fulgidus* Lsm as a reference structure presents fewer residues with a low confidence index than the rest.

The three models generated were statistically validated using Ramachandran diagrams by analyzing the dihedral angles of all the residues of the polypeptide chain and paying attention to the amino acids’ location in the forbidden zones. Of the three models generated, the model using the *Hbt. salinarum* Lsm as the reference structure showed the highest number of amino acids in the forbidden regions. In contrast, the model with the lowest number of amino acids in the prohibited areas was modeled using the Lsm protein from *A. fulgidus* as the reference structure (Figure 2B; Appendix A). Based on all the results obtained, we can state that the model that best fits the Lsm protein of *Hfx. mediterranei* is the model obtained using the Lsm protein of *A. fulgidus* as a reference structure, although when the three models obtained with SWISS-MODEL are compared with the prediction made with AlphaFold2 (model 4), model 1 is the one with the lowest RMSD value (Appendix A, Lsm- AlphaFold2-model).

### 2.2. DNA Microarray Analysis

In previous studies [33], we analyzed the strains HM26 and HM26-Δ*Sm1* under standard and stress conditions, and we explored the growth in two nitrogen sources such as ammonium and nitrate. The Sm1 deletion mutant showed no differences in the defined media in the presence of 40 mM ammonium with the parental strain but presented a more prolonged lag phase in 40 mM nitrate as a nitrogen source. This outcome might indicate that Lsm is involved in regulating nitrogen assimilation. To delve deeper into these results, we chose to study the effect of nutrient deficits such as carbon and nitrogen and to compare the differences between the parental and the mutant strains via microarray.

Transcriptomic expression analysis of the HM26-Δ*Sm1* deletion mutant versus the parental strain (HM26) was performed under 120 h nitrogen starvation and 120 h carbon starvation conditions. The transcriptional differences between nitrogen and carbon starvation in the HM26 parental strain and the HM26-Δ*Sm1* deletion mutant strain were also analyzed, allowing us to investigate the effect of these types of deficiencies in the different strains.

The comparison between the HM26-Δ*Sm1* mutant and the parental strain under carbon source starvation conditions provides information about the differentially expressed genes due to the deletion of Lsm. Firstly, we found very few genes with a significant Log_2_FC, as shown in Figure 3.

Of the up-expressed genes, two are hypothetical proteins and three are genes with known functions, and of the down-expressed genes, three are hypothetical proteins and five are genes with possible known functions. The genes up-expressed in the carbon deficit in HM26-Δ*Sm1* versus the parental HM26 strain are a type IV pilin (HFX_6257), related to cell motility, and two transcriptional regulators: a C2H2-type zinc finger protein (HFX_4107) and a protein of the PadR family (HFX_0688). On the other hand, the genes down-expressed are a CBS domain-containing protein (HFX_1377), whose domains may function as sensors of intracellular metabolites, and recently, it has been reported that CBS domain-containing proteins can sense cell energy levels [50,51]; a twin-arginine translocation signal domain-containing protein (HFX_6411) involved the transport of proteins from the cytoplasm across the inner/cytoplasmic membrane (IM); biotin synthase BioB (HFX_5079), related to biotin synthesis; a thiolase family protein (HFX_6051), involved in the degradation of ketone bodies, fatty acids, amino acids (tryptophan, valine, leucine, isoleucine, and lysine) and benzoate, and the biosynthesis of secondary metabolites; and a high-potential iron-sulfur protein (HFX_6278) related to potential electron acceptors. These results indicate that the Lsm protein, in the absence of a carbon source, mediates the negative regulation of genes related to cell motility and transcriptional regulation and the positive regulation of genes related to biotin synthesis, the synthesis and degradation of fatty acids, amino acids, carbon fixation pathways and the biosynthesis of secondary metabolites (Appendix A).

The comparison between the HM26-Δ*Sm1* mutant and the parental strain under nitrogen source starvation conditions is shown in Figure 4.

Of the 44 down-expressed genes, 7 correspond to hypothetical proteins and 37 genes with known functions. The genes with higher expression in nitrogen starvation in HM26-Δ*Sm1* versus the parental HM26 strain are an aldehyde dehydrogenase (HFX_4023), which could be involved in pathways such as carbon metabolism (glycolysis/gluconeogenesis and pyruvate metabolism), fatty acid degradation, amino acid degradation (valine, leucine, isoleucine, and lysine), glycerolipid metabolism, secondary metabolite biosynthesis, and cofactor biosynthesis; and a beta-ketoacyl-ACP reductase (HFX_1281), related to fatty acid, biotin, and secondary metabolite biosynthesis; acetamidase (HFX_1552) and alkyl sulfatase (HFX_4025), which are not associated with any metabolic pathway in the KEGG database. Most of the genes with lower expression are related to energy metabolism involved with oxidation-reduction processes, specifically with the electron transport chain. In addition, the genes related to gene expression, namely two transcriptional regulators and five genes encoding universal stress proteins (UpsA), are less expressed. These results indicate that the Lsm protein under conditions of nitrogen source deficiency could mediate the negative regulation of genes related to carbon metabolism, fatty acid degradation, amino acid degradation, and biotin biosynthesis and the positive regulation of genes related to energy metabolism, transcriptional regulation, and stress proteins (Appendix A).

The comparison of the parental strain (HM26) in carbon starvation versus nitrogen starvation is shown in Figure 5.

A total of 42% of the up-expressed (40 proteins) and 22% of the down-expressed (32 proteins) genes correspond to hypothetical proteins (Appendix A), resulting in 56 up-expressed (Appendix A) and 116 down-expressed genes (Appendix A) with known functions. The up-regulated genes are mostly related to carbon metabolism (20%), DNA metabolism and processing (20%), amino acid metabolism (18%), and energy metabolism (9%). In addition, genes related to environmental information processing, nitrogen, and lipid metabolism are also more highly expressed. The following enzymes are noteworthy, as they have Log_2_FC values > 5: glutamate dehydrogenase enzyme (HFX_2178), subunit I and II of cytochrome c oxidase enzyme (HFX_1732 and HFX_1733), and DNA starvation/stationary phase protection protein (HFX_1685). On the other hand, the down-regulated genes are mainly related to amino acid metabolism (14.6%), cellular transport (13.7%), carbon metabolism (12%), nitrogen metabolism (10.35%), DNA metabolism and processing (10.35%) and energy metabolism (7%). In addition, genes related to vitamin and cofactor metabolism, nucleotide metabolism, lipid metabolism, environmental information processing, genes encoding several stress proteins (HFX_1094, HFX_0946, HFX_1885, HFX_0369, and HFX_1928), and genes related to signaling and cellular processes are also more highly expressed. The following proteins are noteworthy, as they have Log_2_FC values < −4: stress protein UpsA (HFX_1928); MFS transporter (HFX_2003) and DMT family transporter (HFX_5153); transcriptional regulators (HFX-0453, HFX_0731, and HFX_1124); and enzymes related to nitrogen assimilation: ferredoxin-nitrite reductase (HFX_2005), PII nitrogen regulatory proteins (HFX_0092 and HFX_0094), the enzyme glutamate synthase (HFX_0844) and the ammonium transporter (HFX_0095). In comparing HM26-Δ*Sm1* in carbon starvation versus nitrogen starvation, the same profile as in the parental HM26 strain is observed, except for genes regulated by the Lsm protein, as expected.

Table 1 shows the genes regulated by the Lsm protein depending on the deficiency type. As can be seen, genes related to energy metabolism, stress proteins, and biotin synthesis are highly regulated by the Lsm protein in nitrogen source deficiency but not in carbon source deficiency. All of them present a Log_2_FC < −2, so the Lsm protein mediates the positive regulation in the expression of these genes. On the other hand, in both conditions, the Lsm protein regulates transcriptional regulators. In carbon deficiency, the Lsm protein mediates the negative regulation of the PadR family transcriptional regulator (Log_2_FC = 2.37). In nitrogen deficiency, the Lsm protein mediates the positive regulation of two transcriptional regulators (HFX_1341 and HFX_0165) (Log_2_FC < −2).

As can be seen in Figure 6, and as expected, the genes related to carbon and nitrogen metabolism present a similar pattern in the comparisons “carbon deficiency versus nitrogen deficiency in the parental strain HM26 (HM26HC/HM26HN)” and “carbon starvation versus nitrogen starvation in the HM26-Δ*Sm1* mutant strain (Δ*Sm1*HC/Δ*Sm1*HN)”, except for the genes that are regulated by the Lsm protein.

### 2.3. Overexpression and Purification

Cloning the *lsm* gene into the overexpression vector pTA1992 was successful (Appendix A). Next, chemo-competent *E. coli* JM110 cells were transformed with pTA1992-Lsm-HisTag to obtain unmethylated DNA for the efficient transformation of *Hfx. mediterranei*. The purified pTA1992-Lsm-HisTag plasmids were checked via Sanger sequencing (STAB VIDA, Caparica, Portugal).

The *Hfx. mediterranei* HM26 and HM26-Δ*lsm* strains were transformed with the pTA1992-Lsm-HisTag construct. HM26 and HM26-Δ*lsm* were grown in 10 mL of the complex medium at 42 °C to the stationary phase by collecting aliquots at different times (3 h, 6 h, 12 h, and 24 h), which were analyzed on 14% SDS-PAGE and detected using Coomassie blue staining (Figure 7A). There are no noticeable differences between 12 h and 24 h of growth. As for the HM26 and HM26-Δ*lsm* strains, the overexpression appears slightly higher in the HM26-Δ*lsm* strain. Based on these results, it was decided to perform homologous overexpression of the Lsm protein in the HM26-Δ*lsm* strain in larger culture volumes during 12 h of growth.

After the overexpression of the Lsm protein in the HM26-Δ*lsm* strain for 12 h of growth in 50 mL of complex medium, affinity chromatography was performed using a His-Trap HP 5 mL column (GE Healthcare Life Sciences, Barcelona, Spain). All the elution fractions presented only the purified Lsm protein, so these fractions were pooled and the protein concentration was measured, obtaining a concentration of 0.08 mg/mL in a volume of 30 mL (2.4 mg of Lsm protein) (Figure 7B). After dialyzing the sample, a concentration of 0.06 mg/mL was obtained in a volume of 28 mL (1.68 mg of Lsm protein), and protease cleavage was performed to remove the His-tail of the Lsm protein. Finally, the second affinity chromatography was performed with the same His-Trap HP 5 mL column (GE Healthcare Life Sciences, Barcelona, Spain). In the fractions corresponding to the washes (Figure 7B), the purified Lsm protein was obtained without the His-tag. The fractions from the washes containing the Lsm protein were then pooled and the concentration was measured to obtain 0.032 mg/mL in a volume of 30 mL (0.96 mg of purified Lsm protein).

### 2.4. Molecular Mass Determination of Lsm Protein

The elution volume obtained for each of the standard proteins and the recombinant Lsm proteins is shown in Appendix A. The theoretical molecular weight of the Lsm protein monomer is 8.2 kDa, as shown in Figure 8A, using a concentration of 0.5 M NaCl, 33% of the Lsm protein has a trimer quaternary structure (27.5 kDa) and 66% has a quaternary structure of 6 hexamer rings (36 subunits, 295 kDa); using a concentration of 1 M NaCl, a single peak with a quaternary structure of 6 hexamer rings (36 subunits) is obtained; using a 1.5 M NaCl concentration, 50% of the Lsm protein is obtained as a quaternary structure of 12 hexamer rings (72 subunits, 588.8 kDa), and the other 50% has a quaternary structure of 6 hexamer rings (36 subunits); finally, using a 2 M NaCl concentration, a single peak with a quaternary structure of 6 hexamer rings (36 subunits) is obtained.

All the fractions obtained after gel filtration chromatography in the presence of different NaCl concentrations (0.5 M, 1 M, 1.5 M, and 2 M) of the purified Lsm protein were analyzed via Western blot using an antibody specific for the Lsm protein to verify that they corresponded to this protein (Figure 8B). As can be seen, peaks 1 and 2, corresponding to 12 hexamer rings and 6 hexamer rings, respectively, are formed by the Lsm protein. On the other hand, from the sample of peak 3, corresponding to a trimer, no signal was obtained using this technique, so it cannot be affirmed that the Lsm protein is organized in trimers.

## 3. Discussion

In this work, different bioinformatics tools have been used to increase the knowledge of the Lsm protein in *Hfx. mediterranei.* The species of the class Halobacteria are adapted to grow in salt concentrations higher than 2 M NaCl, employing a strategy (salt-in strategy) that consists of maintaining high intracellular salt concentrations equivalent to or higher than the external concentration of the medium, explicitly accumulating KCl in their cytosol to maintain osmotic equilibrium [52,53]. The proteins have a highly negative surface charge to tolerate these high potassium levels and exhibit a highly acidic proteome [54]. Interestingly, the percentage of similarity between the Lsm protein of *Hfx. mediterranei* and homologous proteins from eukaryotic species is higher than 45%, despite having a high percentage of acidic residues. Moreover, the amino acid sequence of Lsm from *Hfx. mediterranei* corresponds precisely to the consensus amino acid sequence of eukaryotic Lsm proteins [11,33,34], with only a few similarities to the consensus sequence of the bacterial Hfq proteins [55]. In addition, all the RNA-binding residues of the inner and outer pockets of *P. abyssi* [44] are conserved in *Hfx. mediterranei*. These residues are exclusively conserved in the Lsm proteins of Halobacteria class species [34]. The secondary structure of the Lsm protein from *Hfx. mediterranei* was predicted, obtaining an α-helix at the N-terminal end, followed by 6 β-sheets; however, the Lsm proteins are characterized by an α-helix at the N-terminal end, followed by 5 β-sheets [21], so we proposed a more likely hypothetical structure from the results obtained. Although the Lsm protein of *Hbt. salinarum* R1 is the first solved Lsm protein structure of a halophilic microorganism [14], the three-dimensional model that best fits the *Hfx. mediterranei* protein is the Lsm protein of *A. fulgidus*, observing only five residues located in the variable region between the β3 and β4 sheets.

The transcriptional differences between carbon deficiency (120 h) and nitrogen deficiency (120 h) in the parental strain HM26 (R4 Δ*pyrE2*) and in the HM26-Δ*Sm1* mutant strain (R4 Δ*pyrE2* Δ*Sm1*) were analyzed via microarray analysis. The carbon deficiency versus nitrogen deficiency of the HM26 strain showed that among the up-regulated proteins, the most significant differences are the enzyme glutamate dehydrogenase, subunits I and II of the enzyme cytochrome c oxidase, and the stationary/starvation phase DNA protection protein. At the same time, the down-regulated proteins are the stress protein UpsA, MFS, EamA transporter, three transcriptional regulators, and enzymes related to nitrogen assimilation. In addition, under conditions of carbon deficit, enzymes connected to amino acid catabolism and key enzymes of carbon metabolism are more highly expressed, allowing the assimilation of the amino acids, carbon assimilation via gluconeogenesis, and energy production via the Tricarboxylic Acid Cycle (TCA), the electron chain and oxidative phosphorylation. These results agree with the results obtained in *Hfx. volcanii*, where it was found that genes related to the TCA, the electron transport chain, and ATP synthesis are negatively regulated in the presence of glucose [56]. Moreover, as expected, the ammonium in the medium is assimilated by the enzyme glutamate dehydrogenase (Log_2_FC = 7.26), which acts at high ammonium concentrations. In contrast, the enzyme nitrite reductase, PII nitrogen regulatory proteins, and the ammonium transporter are more highly expressed under nitrogen deficit conditions [57,58]. Several genes encoding different transcriptional regulators are also significantly differentially expressed. Based on our results, the transcriptional regulators HFX_0168, HFX_6439, HFX_1280, and HFX_6454 are closely related to gene regulation under carbon starvation. Eight transcriptional regulators are involved in gene regulation under nitrogen-deficient conditions. It is noteworthy that Universal Stress Proteins (USPs) are relevant in nitrogen deficit conditions, since five proteins of this type show Log_2_FC values between −2.63 and −4.03. These proteins are expressed under various conditions, such as heat shock, membrane damage, carbon, nitrogen, sulfate, phosphate, and amino acid starvation [59,60,61,62]. Under stress conditions, USPs are overexpressed via different mechanisms that help the organism to survive [43,63]. Therefore, this experiment has allowed the first identification of USPs in *Hfx. mediterranei* that mediate survival under nitrogen deficit conditions. The contrast between HM26-Δ*Sm1* and HM26 in carbon starvation provides information on the regulation by the Lsm protein under this condition. The results of the contrast HM26-Δ*Sm1* versus the parental strain HM26 in carbon deficiency indicate that the Lsm protein under carbon starvation mediates the negative regulation of proteins related to cell motility (flagellin), DNA metabolism and transcriptional regulation (PadR family) and the positive regulation of the biotin synthase BioB related to biotin synthesis and thiolase related to carbon metabolism, degradation of ketone bodies, fatty acids, amino acids and benzoate, and biosynthesis of secondary metabolites. On the other hand, under nitrogen starvation, the Lsm protein mediates the negative regulation of the enzyme aldehyde dehydrogenase, which is involved in carbon metabolism (glycolysis/gluconeogenesis and pyruvate metabolism), fatty acid degradation, amino acid degradation (valine, leucine, isoleucine, and lysine), glycerolipid metabolism, secondary metabolite biosynthesis, and cofactor biosynthesis; and beta-ketoacyl-ACP reductase, related to fatty acid, biotin, and secondary metabolite biosynthesis. Moreover, the Lsm protein mediates the positive regulation of proteins related to energy metabolism, specifically with the electron transport chain and oxidative phosphorylation. In addition, genes related to gene expression, specifically two transcriptional regulators and five genes encoding universal stress proteins (UpsA), are also positively regulated by Lsm. Although the Hfq proteins have been extensively studied, their function under carbon or nitrogen deficit conditions has never been analyzed. Moreover, only the transcriptome of a deletion mutant of the Sm1 motif of the *lsm* gene in *Hfx. volcanii* has been studied, making this the only transcriptomic study related to the *lsm* gene in archaea. Specifically, this study was performed under carbon-rich conditions, where, interestingly, the Lsm protein mediates the regulation of genes related to cell motility (*flgA1* and *flgA2*) [32].

Homologous overexpression and purification of the *Hfx. mediterranei* Lsm protein has allowed us to determine its quaternary structure by means of molecular exclusion chromatography, obtaining the *Hfx. mediterranei* Lsm protein can form hexameric complexes, which can aggregate into 6 or 12 hexameric rings depending on the NaCl concentration and without RNA. Bacterial Hfq proteins form hexamers [9], whereas eukaryotic Lsm/Sm proteins form heteroheptameric complexes. In contrast, in archaea, the Lsm protein structures are composed of homohexameric [10] or homoheptameric complexes [8,11,12,13,16]. The hexameric or heptameric rings of the Hfq and Sm/Lsm proteins consist of Lsm subunits that assemble in the absence of RNA [8,10,11,12,13,16]. In *A. fulgidus*, the Lsm2 protein was found to form hexamers without RNA but was shown to assemble into stable heptamers upon adding U-rich RNA [10,15]. The Lsm1 structures from *A. fulgidus* and *P. abyssi* revealed that this protein binds U-rich RNA like Hfq and Lsm from eukaryotes, binding inside the proximal face pore, and a U residue binds to each subunit [8,44]. In addition, an unexpected property demonstrated for the Lsm proteins in archaea is their polymerization into long, well-ordered fibers under physiological conditions. These polar fibers are formed through the head-to-tail stacking of heptamers [21,64]. The *E. coli* Hfq protein polymerizes into well-ordered fibers that closely resemble those found in archaea. However, Hfq and Lsm fibers assemble in different ways. At the same time, the Lsm proteins polymerize into polar tubes via head-to-tail stacking, and the Hfq proteins polymerize into helical fibers formed by layers of 36 monomers arranged hierarchically as hexamers [64]. The biological significance of polymerizing these proteins (Lsm and Hfq) is unknown. Still, fiber formation is reconcilable with several physiological functions, such as sequestering mRNA or as a proper Hfq reservoir. Remarkably, (i) polymerization occurs in the absence of RNA and is, therefore, an intrinsic feature of these proteins (not an RNA-induced artifact); (ii) Hfq oligomers can bind RNA, so any physiological functions of Hfq and Sm polymers are probably related to RNA in vivo; and (iii) Hfq polymers can influence DNA packaging [64]. Finally, the Lsm proteins are not the only example of RNA-binding proteins that polymerize into fibers; similar biochemical (RNA binding) and biophysical (polymerization) properties have been described for the HIV Rev protein, forming helical polymers that function to coat viral nucleic acids [65,66,67]. However, single-ring Lsm proteins are considered biologically relevant units, i.e., the hexameric or heptameric complex. Based on our results, we hypothesize that the Lsm protein of *Hfx. mediterranei*, in the absence of RNA, can polymerize into fibrillar structures formed by hexameric rings. However, probably upon the addition of RNA, these hexamers assemble into heptamers, as in *A. fulgidus.* This hypothesis is supported by the results obtained via homology modeling and AlphaFold2 prediction, which shows that the structure is heptameric (Appendix A Lsm_AlphaFold2).

## 4. Materials and Methods

### 4.1. Bioinformatic Analysis

The predictive analysis of the physical and chemical properties of the *Hfx. mediterranei* Lsm protein was already conducted using ProtParam from ExPASy (www.expasy.org, (accessed on 22 January 2022)) and the amino acid sequence o*f Hfx. mediterranei* was aligned and compared to the amino acid sequence of the Lsm protein from *P. abyssi* [33,34].

The secondary structure of the Lsm protein of *Hfx. mediterranei* was predicted using Sequence Annotated by Structure, SAS (v1) [45], and Jpred 4 (v4) [46]. To generate a three-dimensional model of the structure of the Lsm protein of *Hfx. mediterranei*, we first performed an alignment based on the amino acid sequence against the PDB database [47] using the BLAST tool [48] to obtain a suitable reference structure for modeling the Lsm protein. The three reference structures with the highest coverage, sequence identity, and lowest E-value were chosen as reference structures, avoiding using reference structures with mutations that could cause critical structural modifications. For the comparative modeling of the three-dimensional structure, SWISS-MODEL from ExPASy was used [68]. In addition, to check that the generated models were statistically correct, Ramachandran diagrams were obtained for each of the models using the MolProbity web service (http://molprobity.biochem.duke.edu/ (accessed on 6 December 2023)) by observing the dihedral angles of all the residues in the polypeptide chain and paying attention to the location of amino acids in the forbidden zones [69]. Moreover, the prediction of the Lsm protein structure was obtained using the AlphaFold2 software (ColabFold v1.5.5) https://colab.research.google.com/github/sokrypton/ColabFold/blob/main/AlphaFold2.ipynb [49] (accessed on 6 December 2023) and its multimeric models. The algorithm was run through the Google Collaboratory (Colab) notebook template application and servers. The program was run with the default parameters, including six and seven monomers.

### 4.2. Strains, Plasmids, and Culture Conditions

The *Hfx. mediterranei* strains R4 (ATCC 33500^T^) [70], *Hfx. mediterranei* HM26 (R4 Δ*pyrE2*) [71], *Hfx. mediterranei* HM26-Δ*lsm* (R4 Δ*pyrE2* Δ*lsm*) [33] and *Hfx. mediterranei* HM26-Δ*Sm1* (R4 Δ*pyrE2* Δ*Sm1*) [33] were employed in this study. In the HM26-Δ*lsm* deletion mutant, the *lsm* ORF (HFX_2733) was deleted entirely. In contrast, in the HM26-Δ*Sm1* deletion mutant, only the Sm1 motif of the *lsm* gene was deleted and, therefore, contains an intact internal promoter of the *rpl37e* gene located in the Sm2 motif.

The *Hfx. mediterranei* strains were grown in different culture media at 42 °C containing 25% (*w*/*v*) salt water (25% SW) [70]. The pH of all the media was adjusted to 7.2–7.4, with NaOH. To study the transcriptional expression in nitrogen and carbon starvation via DNA microarray, the *Hfx. mediterranei* cells were grown in a defined medium with 40 mM NH_4_Cl and 10 g/L glucose supplemented with 0.0005 g/L FeCl_3_, 0.5 g/L KH_2_PO_4_, 100 mM MOPS, and 50 µg/mL uracil until the mid-exponential phase. Subsequently, the cells were harvested by means of centrifugation for 10 min at 13,000× *g*, washed twice with 25% SW, and transferred to a medium without a nitrogen or carbon source. For the homologous overexpression assay in HM26, complex medium (25% SW, 0.5% (*w*/*v*) yeast extract, and 100 mM MOPS) was used, and aliquots were collected at various times (3 h, 6 h, 12 h, and 24 h).

The *Escherichia coli* strains DH5α for cloning and JM110 for preparing unmethylated DNA for efficient transformation of *Hfx. mediterranei* were grown overnight in Luria–Bertani medium with ampicillin (100 µg/mL) at 37 °C.

The plasmid pTA1992, kindly provided by Dr. Thorsten Allers (University of Nottingham, UK), was used for the protein overexpression. This vector contains pHV2 origin, *pyrE2*, and *hdrB* markers to allow the selection of positive clones on media lacking uracil and thymidine and strong p.syn synthetic promoter for the constitutive overexpression of halophilic proteins with an N-terminal His-tag and/or a C-terminal StrepII-tag [72,73].

### 4.3. DNA Microarray Analysis

The experimental design of the array can be seen in Appendix A. The transcriptional analysis of the HM26-Δ*Sm1* deletion mutant versus the parental strain HM26 was performed under nitrogen and carbon starvation for 120 h. Following the product specifications, the total RNA was isolated from these samples using the RNeasy Mini Kit (Qiagen, Hilden, Germany). The quality and quantity of the total RNA were checked with a Bioanalyzer (Agilent Technologies, Santa Clara, CA, USA) and NanoDrop (Thermo Fisher Scientific, Waltham, MA, USA), respectively. All the samples showed an RNA integrity number (RIN) higher than 7.

The transcriptional analysis using the expression microarray technique was designed based on the *Hfx. mediterranei* genome using eArray software (v. 6.1, Agilent Technologies, CA, USA) [57]. The microarray analysis was conducted at Bioarray, S.L. Company (Alicante, Spain). Three probes were designed for each gene, all with a length of 60 nucleotides. The cDNA labeling was performed with the two-color microarray-based gene expression analysis v. 6.5 (Agilent Technologies, CA, USA) according to the manufacturer’s protocol. This consisted of labeling the reference condition and the condition of interest with a different fluorochrome (Cy3 and Cy5, respectively). An oligo array hybridization buffer was added, and the samples were applied to the microarray in an Agilent SureHyb-enabled hybridization chamber to perform the hybridization. The microarray images were acquired using a G2505C scanner (Agilent Technologies, CA, USA) and analyzed with Agilent Feature Extraction Software v. 10.7.3.1 (Agilent Technologies, CA, USA).

All the statistical and differential expression analyses were performed using Bioconductor’s Limma package (http://www.bioconductor.org/ accessed 5 June 2021). Gene expression was considered up- or down-expressed if the Log_2_FC of the fold change was ≥2.0-fold (up-expressed) or ≤−2.0-fold (down-expressed) and statistically significant (*p*-value < 0.05). The microarray data can be accessed from the Gene Expression Omnibus (GEO) database (GSE244624).

### 4.4. Homologous Overexpression of lsm in Hfx. mediterranei HM26

The *lsm* gene was amplified via PCR from the genomic DNA of *Hfx. mediterranei* R4 (100 ng DNA, 1X buffer, 2.5 mM MgCl_2_, 0.2 mM dNTPs, 100 pmol/oligonucleotide, and 1 U of Supreme NZYTaqII DNA Polymerase (NZYTech, Lisbon, Portugal)) using the direct primer pTA1992Lsm_D (5′-CAC CAC CAC CAC CAC CAC CAC CAC CTT GAA GTC CTC TCT TTC AGG GAC CCA TGA GCG GCC GAC CCG ACC CCT CGA-3′) and the reverse primer pTA1992Lsm_R (5′-CTG CGG CGG CCG CAA GCT TTT TCA TGC TTT GAT GGT CAC GAC GAC GTT ATC-3′). The primers were designed according to the instructions of the In-Fusion**^®^** HD Cloning kit (Clontech, Torrejón de Ardoz, Madrid, Spain). The PCR product was purified using the Gel Band Purification Kit (GE Healthcare Life Sciences, Barcelona, Spain) according to the manufacturer’s instructions. The pTA1992 vector (1 µg) was digested with the *Pci*I and *EcoR*I restriction enzymes (Thermo Fisher Scientific, Waltham, MA, USA) at 37 °C for 1 h.

Once the pTA1992 vector was linearized and the amplified insert was purified, cloning was performed using the In-Fusion**^®^** HD Cloning kit following the manufacturer’s instructions, transforming chemo-competent *E. coli* DH5α cells via the standard heat shock transformation protocol [74] using LB with Amp (100 µg/mL) as the selection medium. Once the plasmids were verified, chemo-competent *E. coli* JM110 cells were transformed with the plasmids [74] to prepare unmethylated DNA for the efficient transformation of *Hfx. mediterranei*. Similarly, pTA1992-*lsm*-HisTag plasmids transformed into *E. coli* JM110 were purified with the E.Z.N.A.**^®^** Plasmid DNA Mini Kit I (Omega Bio-Tek, Norcross, GA, USA). Finally, the pTA1992-*lsm*-HisTag plasmid sequence was checked by means of Sanger sequencing (STAB VIDA, Caparica, Portugal).

The *Hfx. mediterranei* strains HM26 and HM26-Δ*lsm* were transformed with the pTA1992-*lsm*-HisTag [71] and selected on agar plates with a defined medium (25% SW, 0.25% (*w*/*v*) Casamino acids, 10 mM NH_4_Cl, 4 mM CaCl_2_, 0.5 g/L K_2_HPO_4_, 0.0005 g/L FeCl_3_ and 100 mM MOPS). The plates were incubated at 42 °C for 7–10 days until pink colonies were visible.

The colonies transformed with the pTA1992-*lsm*-HisTag of HM26 and HM26-Δ*lsm* strains were grown in 10 mL of the complex medium at 42 °C to the stationary phase by collecting 1 mL aliquots at various times (3 h, 6 h, 12 h, and 24 h). The different aliquots were harvested via centrifugation at 13,000× *g* for 30 min and resuspended in 40% (*w*/*v*) binding buffer (20 mM Tris-HCl, 1.5 M NaCl, 20 mM imidazole, pH 7.4). The cells were lysed in the cold via sonication (9 pulses of 40 s, at 10% amplitude) using a Digital Sonifier S450D sonicator (BRANSON, Barcelona, Spain). The cell lysate was centrifuged at 16,000× *g* for 15 min at 4 °C, and the supernatant was collected to obtain the total protein extract. All the extracts obtained at different incubation times were analyzed on 14% SDS-PAGE using PageRuler Plus Prestained Protein Ladder (Thermo Fisher Scientific, Waltham, MA, USA) as molecular weight markers. Proteins were detected via staining with Coomassie Brilliant Blue.

### 4.5. Protein Purification

Based on the results obtained, it was decided to overexpress the Lsm protein from the HM26-Δ*lsm* strain after 12 h of growth in 50 mL of the complex medium at 42 °C. After growth, the culture was centrifuged at 13,000× *g* for 30 min and resuspended in binding buffer (20 mM Tris-HCl, 1.5 M NaCl, 20 mM imidazole, pH 7.4). The total protein extract was obtained in the same way as in Section 2.4. Finally, the extract was filtered through a 0.45 µm filter for affinity chromatography.

Once the extract was prepared, affinity chromatography was performed using a 5 mL His-Trap HP column (GE Healthcare Life Sciences, Barcelona, Spain). A flow rate of 5 mL/min was used during all the chromatographic steps. The column was equilibrated with 5 CVs (column volumes) of binding buffer (20 mM Tris-HCl, 1.5 M NaCl, 20 mM imidazole, pH 7.4), and a 40 mL sample volume was applied (maximum recommended by the manufacturer). The column was washed with binding buffer (10 CV). Finally, the protein was eluted with 7 CV of elution buffer (20 mM Tris-HCl, 1.5 M NaCl, 500 mM imidazole, pH 7.5). The concentration and quality of the elution fractions were visualized using 14% SDS-PAGE denaturing electrophoresis, and the protein concentration of each fraction was measured using the Bradford method [75].

Before the cleavage of the His-tag with HRV3C protease, dialysis was performed to remove excess imidazole against 2 L of binding buffer (20 mM Tris-HCl, 1.5 M NaCl, 20 mM imidazole, pH 7.4). Finally, the protein concentration was measured using the Bradford method [75]. Protease cleavage was performed by adding 14 mM β-mercaptoethanol and 1 mg HRV3C protease/100 mg protein to the dialysate solution and incubating for 2 h at room temperature.

The affinity chromatography was performed using a His-Trap HP 5 mL column (GE Healthcare Life Sciences, Barcelona, Spain) following the same protocol as previously explained. In this case, purified Lsm protein is expected to be obtained in the washes after sample application. The concentration and quality of the elution fractions were analyzed, as explained above.

### 4.6. Determination of Molecular Mass

A HiPrep 16/60 Sephacryl S-200 High-Resolution HiPrep column (Cytiva, Barcelona, Spain) was employed to determine the size of the Lsm protein. Before the sample application, the column was equilibrated with 2 CV. The effect of NaCl on the Lsm quaternary structure was analyzed using different buffers with an increasing NaCl concentration (20 mM Tris-HCl, pH 7.5, 0.5 M, 1 M, 1.5 M, or 2 M). Protein standards were loaded to the column using the maximum volume recommended by the manufacturer and a 0.5 mL/min flow rate during all the chromatographic steps. The chromatography was performed with 1 CV using the same buffer with which the column was equilibrated. Finally, the protein concentration was determined via the Bradford method [75] from the elution peaks of the standard proteins and the Lsm protein.

All the peak fractions obtained after performing molecular size-exclusion chromatography at different NaCl concentrations (0.5 M, 1 M, 1.5 M, and 2 M) of the purified Lsm protein were analyzed on 14% SDS-PAGE, as described above.

### 4.7. Western Blot

To verify that the peaks obtained corresponded to the Lsm protein, Western blot was performed, as described in the Western Blotting Principles and Methods Manual (GE Healthcare Life Sciences, Barcelona, Spain), using 10 μg of Lsm protein, rabbit polyclonal anti-Lsm antibody (GenScript, Piscataway, NJ, USA) as the primary antibody at a concentration of 0.6 mg/mL and a 1:50,000 peroxidase-labeled secondary antibody (Thermo Fisher Scientific, Waltham, MA, USA). Immunodetection was performed using the chemiluminescent substrate luminol according to the manufacturer’s instructions with an Amersham^TM^ ECL^TM^ Prime Western Blotting Detection Reagent Kit (GE Healthcare Life Sciences, Barcelona, Spain).

## Figures and Tables

**Figure 1 ijms-25-00580-f001:**
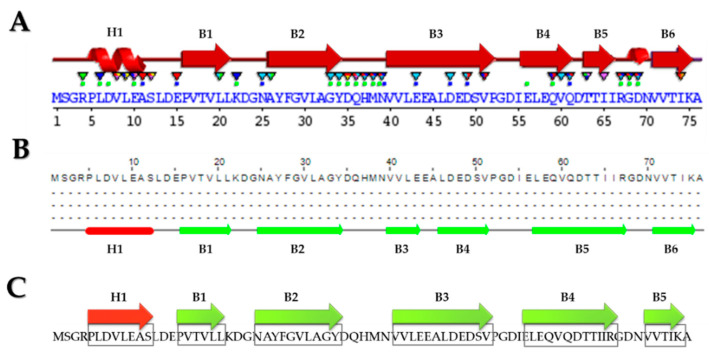
Analysis of the secondary structure of the Lsm protein of *Hfx. mediterranei*. (**A**) Secondary structure by means of SAS. The α-helix structure is represented by a spiral and the β-sheet by an arrow. Green dots indicate residues interacting with RNA/DNA, and blue dots indicate interactions with metals. Triangles indicate active residues deposited on other proteins in PDB. (**B**) Secondary structure by means of Jpred 4. (**C**) Proposed secondary structure. The α-helix structure (H1) is shown in red, and the β-sheet (B1–B6) is green.

**Figure 2 ijms-25-00580-f002:**
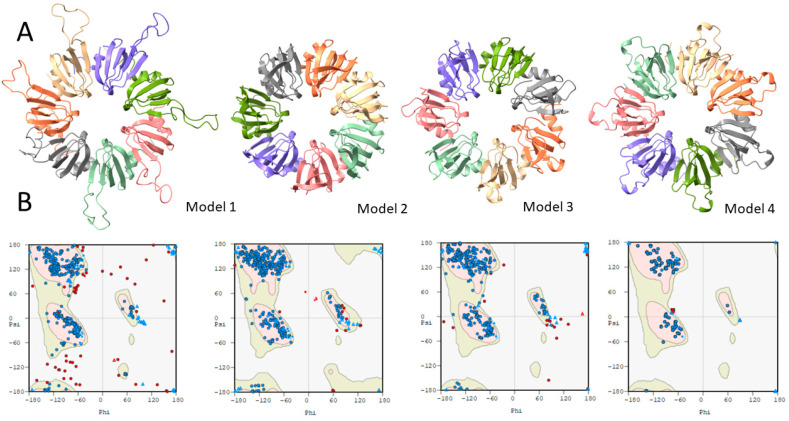
Homology modeling results of the Lsm protein of *Hfx. mediterranei*. (**A**) Structural models obtained via sequence homology from *Hbt. salinarum* (model 1), *A. fulgidus* (model 2) and *P. abyssi* (model 3) with SWISS_MODEL. Model 4 was produced using Colab AlphaFold2. The program was run with the default parameters, including six and seven monomers. The top structures were relaxed using amber [49]. (**B**) Ramachandran plots.

**Figure 3 ijms-25-00580-f003:**
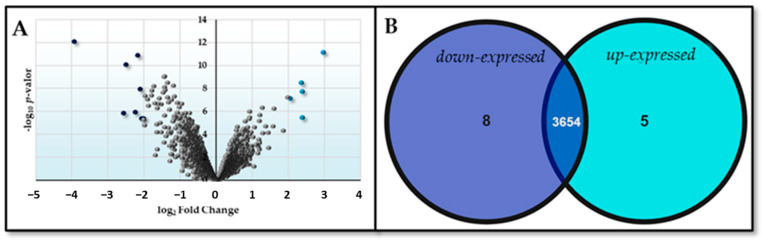
Differential expression represented by (**A**) Volcano plot (dark blue dots indicate down-expressed genes and light blue dots indicate up-expressed genes) and (**B**) Venn diagram of the contrast HM26-Δ*Sm1* in carbon deficiency versus HM26 in carbon deficiency (Δ*Sm1*HC-HM26HC).

**Figure 4 ijms-25-00580-f004:**
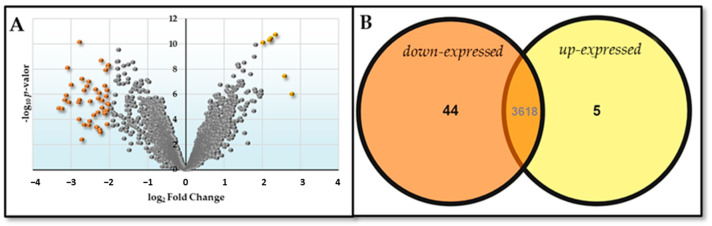
Differential expression represented by (**A**) Volcano plot (orange dots indicate dow-expressed genes and yellow dots indicate up-expressed genes) and (**B**) Venn diagram of the contrast HM26-Δ*Sm1* in nitrogen deficiency versus HM26 in nitrogen deficiency (Δ*Sm1*HN-HM26HN).

**Figure 5 ijms-25-00580-f005:**
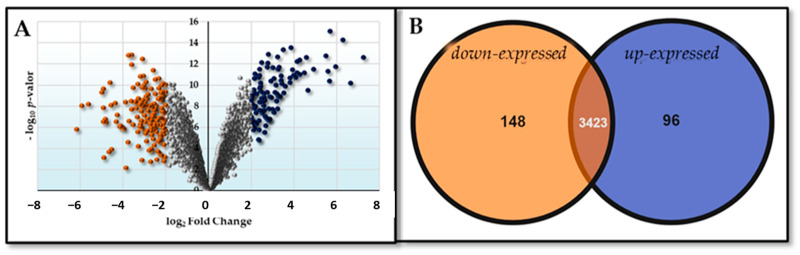
Differential expression represented by (**A**) volcano plot (orange dots indicate dow-expressed genes and dark blue dots indicate up-expressed genes) and (**B**) Venn diagram of the contrast HM26 in carbon deficiency versus HM26 in nitrogen deficiency (HM26HC-HM26HN).

**Figure 6 ijms-25-00580-f006:**
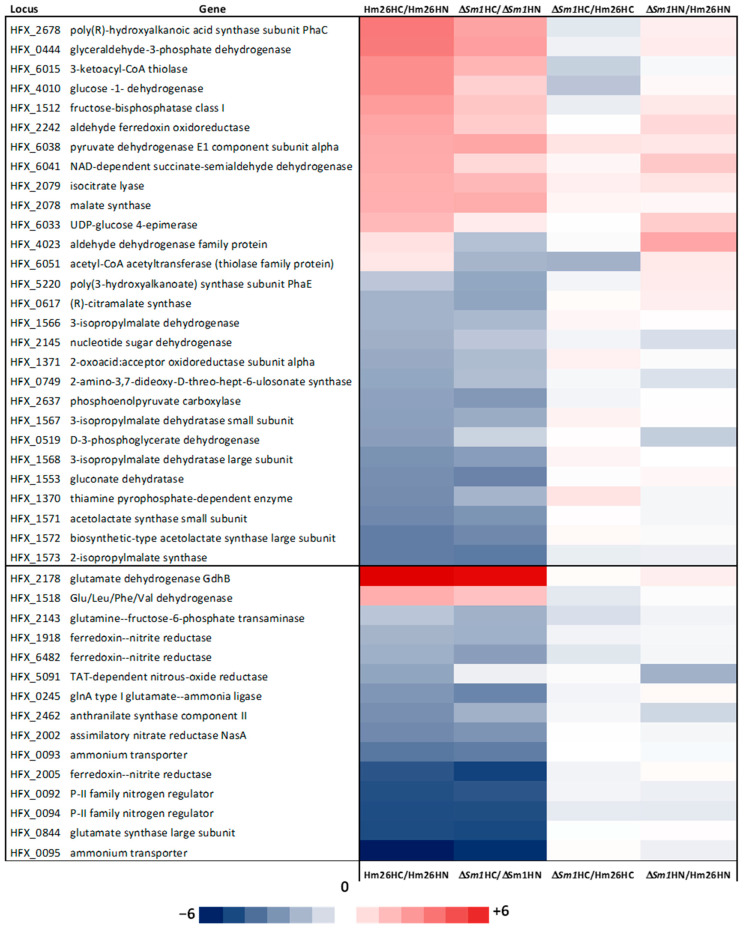
Heat map of the genes related to carbon metabolism (**top**) and nitrogen metabolism (**bottom**). HC: carbon deficit; HN: nitrogen deficit.

**Figure 7 ijms-25-00580-f007:**
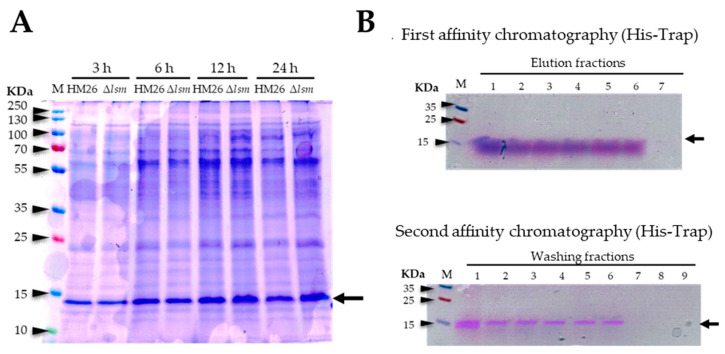
Homologous overexpression and purification of the Lsm protein. (**A**) The HM26 and HM26-Δ*lsm* strains after different growth times (3 h, 6 h, 12 h, and 24 h). (**B**) The fractions obtained after purification of the Lsm protein via affinity chromatography (His-Trap). M: PageRuler Plus Prestained Protein Ladder (Thermo Fisher Scientific, Waltham, MA, USA). The black arrow indicates the size of the Lsm protein of *Hfx. mediterranei*.

**Figure 8 ijms-25-00580-f008:**
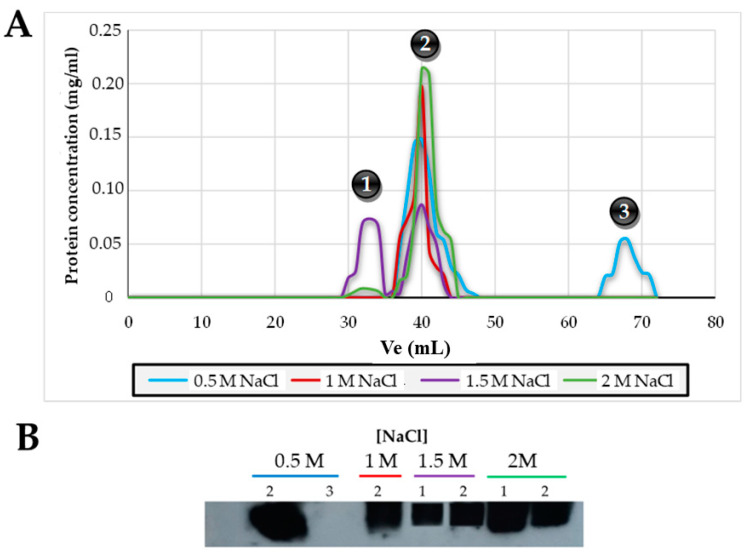
Determination of the Lsm protein structure via molecular exclusion chromatography. (**A**) Chromatograms of the molecular exclusion chromatographies using buffers with different NaCl concentrations. (**B**) Western blot using an antibody specific for the Lsm protein of peaks 1, 2, and 3 of section A. Note: The complete photo of the Western blot is deposited in the Appendix A (Photo_8B).

**Table 1 ijms-25-00580-t001:** Comparison of the genes regulated by the Lsm protein according to the carbon or nitrogen source deficit. C deficit (Log_2_FC for the comparison HM26-Δ*Sm1* vs. HM26 under carbon deficit conditions); N deficit (Log_2_FC for the comparison HM26-Δ*Sm1* vs. HM26 under nitrogen deficiency conditions).

		**C Deficit**	**N Deficit**
**Energy metabolism**			
HFX_0944	b(o/a)3-type cytochrome-c oxidase subunit 1	−0.41	−2.07
HFX_0429	cytochrome ubiquinol oxidase subunit I	−0.33	−2.34
HFX_1925	FAD-dependent oxidoreductase	−0.34	−2.55
HFX_0428	cytochrome d ubiquinol oxidase subunit II	−0.77	−2.72
HFX_0943	cytochrome c oxidase subunit II	−0.89	−2.82
HFX_1927	electron transfer flavoprotein subunit beta/FixA family protein	−0.49	−3.14
HFX_1926	electron transfer flavoprotein subunit alpha/FixB family protein	−0.53	−3.33
**Genes that encode stress proteins**			
HFX_0946	universal stress protein	0.22	−2.05
HFX_1094	universal stress protein	−0.94	−2.09
HFX_2288	universal stress protein	−1.55	−2.13
HFX_1885	universal stress protein	−0.40	−3.10
HFX_1928	universal stress protein	−0.11	−3.24
**DNA metabolism and processing**			
HFX_0688	PadR family transcriptional regulator	2.37	0.29
HFX_4107	C2H2-type zinc finger protein	2.40	0.18
HFX_1341	helix-turn-helix domain-containing protein	0.14	−2.06
HFX_0165	winged helix-turn-helix transcriptional regulator	0.15	−2.42
**Carbon metabolism**			
HFX_4023	aldehyde dehydrogenase family protein	−0.09	2.58
HFX_6051	thiolase family protein	−2.16	0.66
**Metabolism of cofactors and vitamins**			
HFX_5076	dethiobiotin synthase	−0.52	−2.22
HFX_5079	biotin synthase BioB	−1.98	−2.66
HFX_5077	8-amino-7-oxononanoate synthase	−1.39	−2.73
**Nitrogen metabolism**			
HFX_5091	TAT-dependent nitrous-oxide reductase	−0.09	−2.19
**Lipid metabolism**			
HFX_1281	beta-ketoacyl-ACP reductase	−0.47	2.22
**Cell motility proteins**			
HFX_6257	type IV pilin N-terminal domain-containing protein	2.98	−1.80
**Transporters**			
HFX_1359	substrate-binding domain-containing protein	−1.68	−2.26
**Environmental signal processing**			
HFX_5231	CBS domain-containing protein	−1.63	−3.07

## Data Availability

All data are available on reasonable request.

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
