# Peer review of "Analysis of Lsm Protein-Mediated Regulation in the Haloarchaeon Haloferax mediterranei"

_ijms, 2024, doi:10.3390/ijms25010580_

Round 1
Reviewer 1 Report
Comments and Suggestions for Authors
The authors analyzed the Lsm protein of Haloferax mediterranei, with a
focus on oligomeric complex formation of the homologously overexpressed
protein and effects of a deletion mutant on gene expression.
Major parts of the study provide convincing data with conclusive
interpretation. However, in some parts data are insufficiently
documented. In other parts, the interpretation should be reconsidered.
In addition, there are two "terminology" issues where clarification
is required.
Point A, a terminology issue
In the abstract the authors state "The structural bioinformatic analysis
of the Lsm protein has been carried out". This raised the expectation
that a 3D structure has been obtained experimentally and is reported in
the manuscript. However, this is not the case. The authors concentrate
on two aspects where they report and integrate data from structure
prediction (2D structures and 3D structures) without any experimental data.
They provide experimental data related to the number of subunits in
oligomeric complexes, but there are issues with these (see Point B).
Point B, insufficiently documented data
An important aspect are the number of subunits upon formation of an
oligomeric complex. For this analysis, the protein has been homologously
overexpressed, a nice and well-documented experiment. The protein is
then analyzed via molecular exclusion chromatography. Results are
shown in Fig.4. The authors state "The elution volume obtained for each
of the standard proteins and the recombinant Lsm proteins is shown in
Table S3 and Figure S2." There is no Table S3 in the supplementary material
provided by the authors. Figure S2 exists but has no legend. The correlation
coefficient is given as 0.9834. The authors state that the Lsm has
"trimer quaternary structure" and "a quaternary structure of 6 hexamer
rings (36 subunits)". The initial result from molecular exclusion
chromatography is an elution volume, which (based on standards) can be
converted to a total molecular weight. Subsequently, that can be interpreted
in terms of quaternary structure. I did not find any indication of the
total molecular weight in the manuscript (in the absence of Suppl.Table S3).
From the correlation coefficient, in combination with the peak width, the
uncertainty of the total molecular weight estimate needs to be computed.
I would doubt that the uncertainty is below the mass of a single subunit.
The implications of the obtained uncertainty estimate on the result
interpretation needs to be addressed by the authors.
Point C, a terminology issue
The authors use RefSeq-assigned tags (HFX_RS0000-style) throughout their
manuscript. It seems preferrable that they would use the originally
assigned HFX_0000-style ordered locus tags instead (or provide a
correlation table). The HFX_0000-style ordered locus tags have the
advantage to be ordered and permanent. That is not guaranteed for
HFX_RS0000-style tags. As an example, the authors mention HFX_RS00445,
which is only retrieved at NCBI, searching all databases, for Structure,
not for Gene (0 hits) or Protein (0 hits). When NCBI is searched
specifically for Gene, there is the information that "This record
was discontinued". Without pre-knowledge, which of the many NCBI
databases has to be searched, there is no possibility to retrieve
that HFX_RS number. Thus, readers have major difficulties to identify
the proteins/genes which are discussed in the paper. With the originally
assigned ordered locus tag HFX_0092, everything is detected easily
(1 hit in Gene, 2 in Protein, and 2 in Nucleotide). In the case of
HFX_0092, RefSeq has assigned a new tag, HFX_RS20245, which is not
an ordered one.
Point D, interpretation issues
The authors mention regulation of biological processes according to
KEGG maps to which KEGG has assigned the corresponding regulated proteins.
It might be debated if a single regulated gene can affect all of that
biological process (consisting of dozens of genes). This becomes even
more problematic for genes which catalyze one of the reactions of
central intermediary metabolism. Those are represented on multiple
KEGG maps.
One such case is thiolase (HFX_RS17055, HFX_6051), "involved in the
degradation of ketone bodies, fatty acids, amino acids (tryptophan,
valine, leucine, isoleucine, and lysine) and benzoate, and biosynthesis
of secondary metabolites". That's an impressive list, but it can be
questioned if the regulation of that single gene will drastically affect
such a major fraction of metabolism. This is even more questionable
because a total of 5 paralogs assigned to this reaction are encoded
in the genome (HFX_2006, HFX_6015, HFX_6051, HFX_6356, HFX_6358).
The other example is HFX_RS14985 (HFX_4023), an aldehyde dehydrogenase.
It seems unresolved if this enzyme is specfic for acetaldehyde (as the
name implies) or has another aldehyde as its substrate. Of course, it
might also be promiscous, acting on multiple aldehydes in a rather
unspecific was. The best SwissProt hit to HFX_4023 (42% seq_id) is
Bacillus subtilis dhaS which has been experimentally characterized
(PMID:25409630) and has a preference for 3-hydroxypropionaldehyde.
There are 3 paralogs with that annotation in the genome (HFX_1199,
HFX_4023, HFX_6374), and regulation of a single paralog might thus
not have a big effect on metabolism. For this enzyme, the authors
list 6 biological processes ("which is involved in carbon metabolism
(glycolysis/gluconeogenesis and pyruvate metabolism), fatty acid
degradation, amino acid degradation (valine, leucine, isoleucine,
and lysine), glycerolipid metabolism, secondary metabolite biosynthesis,
and cofactor biosynthesis"). However, the various KEGG maps assign
this enzyme to a highly diverse set of reactions (glycolysis/gluconeogenesis
and pyruvate metabolism: conversion of acetaldehyde to acetate;
fatty acid degradation: interconversion of a fatty acid with its aldehyde;
Val/Leu/Ile degradation: conversion of methylmalonate semialdehyde to
methylmalonate; Lys degradation; conversion of trimethylammoniobutanal to
trimethylammoniobutanoate; glycerolipid metabolism: D-glycerate to
D-glyceraldehyde). Half a dozen additional enzymatic reactions are assigned
by KEGG to this same enzyme, listed in Arg/Pro, in His, and Trp metabolism,
in beta-alanin metabolism/CoA biosynthesis. Those are skipped by the authors.
It seems highly unlikely that this single enzyme specifically acts on this
plethora of substrates. This is one of the "KEGG traps", the assignment of
a protein which has only been generally characterized (it contains an
aldehyde dehydrogenase domain) to multiple highly specific reactions.
To list all of those KEGG assignments, as if the enzyme would be the
key player of metabolism, seems highly misleading.
Point E, a statement that may be seen as misleading
The authors modelled the structure by homology to Hbt. salinarum,
A. fulgidus and P. abyssi. The differences to Hbt. salinarum are
unexpectedly large (visual impression from Fig.2). The authors also
mention "the highest number of amino acids in the forbidden regions".
This result is insofar very astonishing as the authors point to
"the highest coverage (98 %), and sequence identity (62.67 %)".
But that statement might be misleading as it seems to refer to the
coverage of the Hbt.salinarum residues in the Hfx.mediterranei sequence.
Coverage is much lower the other way round. Upon BLASTp alignment,
residues 45-60 of Hfx.mediterranei have no counterpart in Hbt.salinarum.
This corresponds to a coverage of just 80% of the Hfx.mediterranei
residues in the Hbt.salinarum sequence, which may well explain the
suboptimal structural modelling results.
Point F: non-tabulated proteins mentioned as regulated in the text
In line 170, HFX_RS06725, a CBS domain protein is mentioned, stating
that the function is unknown, but this protein is not listed in any
table. Another CBS domain protein is listed in Tab.2 (HFX_RS16515,
HFX_5231). It is named a Signal transduction protein for unclear
reasons.
In lines 220 to 235, many distinct and highly regulated proteins are
mentioned. These might also be listed in a (supplementary) table.
Also, a number of regulated "hypothetical proteins" are mentioned but
they seem not to be listed anywhere. These are another candidate set
for a supplementary table.
In the discussion (line 343) four transcription regulators are claimed
to be related to gene regulation under carbon starvation. They seem
not to be mentioned anywhere in the Results section.
Point G: miscellaneous
In line 227-228, after "genes encoding several stress proteins", is
a list of genes, nearly all of which code for "Universal stress protein".
One, HFX_RS13705 (HFX_1958) is an Aldo/keto reductase which, for unclear
reasons, is classified as "Environmental information processing under
stressful conditions". What is the basis for this statement?
Side aspect: the word "y" in line 228 seems Spanish.
Author Response
Please see the attacment

Reviewer 2 Report
Comments and Suggestions for Authors
The manuscript presents interesting information about Lsm protein in Haloferax mediterranei, however, this reviewer opinion is that the manuscript requires major remodeling. A careful reading and reorganization of the manuscript is suggested, with emphasis on clarifying and better presenting the data and ideas.
Abstract:
I believe this abstract HAS to be improved to showcase better the findings of the authors. It is too similar to the end of the paper introduction. And that is not the goal of an abstract.
“ In this work, the haloarchaeon Haloferax mediterranei has been used as a model microorganism”
Why? Here and in the introduction, I believe this case should be better clarified.
“The structural bioinformatic analysis of the Lsm protein has been carried out, and the structural experimental study has been performed by homologous overexpression and purification by molecular exclusion chromatography, obtaining that they can form hexameric complexes, which can aggregate into 6 or 12 hexameric rings depending on the NaCl concentration”
This sentence is way too long. The reader cannot understand and it is a fundamental part of the results.
“The structural bioinformatic analysis of the Lsm protein has been carried out “= and what happened? The abstract is not an introduction to what was done, but has to show the reader what are the findings of the paper.
In a separate sentence, state what happened experimentally.
“In addition, the study of transcriptional expression by microarrays has allowed us to obtain the target genes regulated by the Lsm protein under conditions of nutritional stress: nitrogen or carbon starvation.”
And what were the main findings?
Introduction
Although important information are presented and references are appropriately cited, I believe that, in general, the language and presentation can be improved. I believe the authors should work on better connecting the ideas and contrasting when necessary. Sometimes the use of “contrasting prepositions” is not clear, what are the ideas being contrasted. I suggest a throughout proofreading to improve readability of the introduction to clarify the ideas to the reader.
Lsm or SmAP1 binding to RNAs has been recently mapped in another haloarchaea, Halobacterium salinarum as one of the mechanisms of post-transcriptional regulation (Lorenzetti et al, 2023, mSystems).
Results
In silico analysis at the secondary structure level
For this reviewer, the way the proposed structure was achieved seemed a little arbitrary.
The colors of the triangles, what do they mean?
Homology modeling
Did the authors try to use AlphaFold? They can access its prediction directly from Uniprot. How does that compare to the models obtained by the authors using SWISS-MODEL? I think this is an important comparison to be made and discussed.
“obtaining the parameters in Table S1”
“In contrast, the model with the lowest number of amino acids in the prohibited areas was modeled using the Lsm protein from A. fulgidus as the reference structure (Figure 2B; Table S2).”
this reviewer could not access Table S1 or Table S2
This reviewer is not an expert in protein structure analysis or homology modeling. However, in my opinion the data presented could be either further discussed and explained. I was not fully convinced by the data, the way it was presented.
Hbt salinarum based structure have something in the inner core of the protein. Are these loops from the protein? And the scale of this figure seems different from the other two.
DNA microarray analysis
I believe a short introduction of why these conditions were chosen, based on reference 34 would be very useful for the reader to understand the point of the experiments.
It is difficult for the reader to follow these results.
Have the authors performed a gene enrichment analysis?
Could they present the data more visually, such as a plot with categories of genes?
“The comparison between the HM26-ΔSm1 mutant and the parental strain under carbon source starvation conditions provides information on the regulation of the Lsm protein”
In my opinion, this sentence should be rephrased. When we compare a mutant with the parental strain, we can see the consequences of the deletion. They can be either direct or indirect. I don't think we can say “the regulation of the Lsm protein”, it might give a misguided impression. We are detecting the differentially expressed genes due to the deletion of Lsm.
The first paragraph of this section essentially describes the table below the text. There is no need to do that. I think the authors can give the numbers and general ideas or contexts for genes that will be discussed, but there is no point in describing the table in the text. They should either refer to the table or make a graphic representation of the categories.
The authors could present the data in a Venn diagram, with the number of genes in each comparison, the ones in common and the ones exclusive for each stress. It would be easier to have a better overview of the consequences of Lsm deletion in the different stresses in terms of number of affected genes and also, in the categories of genes that were affected.
Other results:
I could not find Table S3
2.4. Molecular mass determination of Lsm protein
Figure 4: can the author provide the original image for the blot? it is not provided in the file.
The one in figure 4 is cut at the top, cutting the band. I would like to see the whole band.
Discussion
“In addition, under conditions of carbon deficit, enzymes related to amino acid catabolism and key enzymes of carbon metabolism are more highly expressed, allowing the assimi- lation of the amino acids and key enzymes of carbon metabolism are more highly ex- pressed, allowing carbon assimilation via gluconeogenesis and energy production via the TCA (Tricarboxylic Acid Cycle), the electron chain and oxidative phosphorylation”
I suggest revising sentences such as the one above, that are way too long and difficult to understand.
Materials and methods
“The predictive analysis of physical and chemical properties of Hfx. mediterranei Lsm protein was already carried out” = already? it was done before in the references or in this work?
Comments on the Quality of English Language
The sentences are sometimes too long. Constrasting prepositions are sometimes used without constrasting ideas.
Author Response
Please see the atachment
